# Evaluation of Nanoparticle-Based Plasma Enrichment on Individuals with Primary and Metastatic Pancreatic Cancer

**DOI:** 10.3390/cancers17233765

**Published:** 2025-11-25

**Authors:** Ching-Seng Ang, Nicholas A. Williamson, Chelsea Dumesny, Michael G. Leeming, Keshava Datta, Swati Varshney, Mehrdad Nikfarjam, Hong He

**Affiliations:** 1Mass Spectrometry and Proteomics Facility, Bio21 Molecular Science and Biotechnology Institute, University of Melbourne, Parkville, VIC 3052, Australia; nawill@unimelb.edu.au (N.A.W.); leeming.m@unimelb.edu.au (M.G.L.); swati.varshney@unimelb.edu.au (S.V.); 2Department of Surgery, Austin Precinct, University of Melbourne, Heidelberg, VIC 3084, Australia; watsoncj@unimelb.edu.au (C.D.); m.nikfarjam@unimelb.edu.au (M.N.)

**Keywords:** pancreatic cancer, nanoparticles enrichment, plasma proteome, Orbitrap Astral, markers, ADH, ribosomal proteins

## Abstract

Nanoparticle-based plasma enrichment followed by mass spectrometry analysis is a powerful technology to increase the depth of identification of potential protein markers from plasma samples. In a small-scale pilot test, we have shown a 7× increase in the depth of protein identification using nanoparticle-based enrichment as compared to neat plasma digest. We also showed a large increase in abundance of ribosomal proteins and 15 cell surface or secreted cancer-associated proteins in individuals with metastatic pancreatic cancer, showing the usefulness of this technique for future large-scale discovery and validation experiments.

## 1. Introduction

Plasma has been the biological matrix of choice for preventive screening to assess patients’ risk for certain conditions as well as diagnosis of disease and infection. Sampling plasma is inexpensive and minimally invasive, providing ready access to proteins, lipids, and other metabolic products that can be assayed to screen for certain disease states. Circulating proteins can therefore be potential biomarkers for diagnosing various diseases, including cancer. For example, CA 19-9 is an FDA-approved tumor marker for the diagnosis and management of pancreatic cancer (PC) [1]. In an asymptomatic population, however, this biomarker has a positive predictive value for PC below 1 percent, and there is therefore an urgent need to discover more specific biomarkers as a screening method [2]. Human plasma may contain other circulating proteins that are cancer-related and possibly tissue leakage proteins.

Despite being a convenient biological matrix, mass spectrometric analysis of plasma has been historically challenging. The large dynamic range of proteins has long been recognized as an issue for the detection of lower-abundant circulating proteins [3,4] as the most abundant 22 plasma proteins can account for more than 99% of all proteins. Different strategies, such as selective depletion of high-abundant proteins [5,6,7], prefractionation of plasma [8,9], and selective enrichment [10], have been attempted to reduce sample complexity and address the dynamic range issue with limited success. Recently, nontraditional techniques such as affinity-based SomaScan aptamer (short oligonucleotides) assays (SomaLogic, Boulder, CO, USA [11]) and antibody-based proximity extension assays (Olink, Waltham, MA, USA [12]) have emerged and can simultaneously detect and quantify thousands of plasma proteins.

Another emerging technique is selective enrichment of plasma proteins through nanoparticles. Protein coronas form on the surfaces of nanoparticles through non-covalent interactions when introduced into physiological fluids [13,14]. The biomolecular corona consists of a tightly bound inner layer (hard) and an outer layer of loosely associated (soft) proteins that can be removed through washing and centrifugation [15]. The shape, size, surface charge, and surface chemistry significantly influence the physiochemical properties of the particle and hence their ability to bind proteins [14,16,17]. This unique property has been exploited to enrich and allow parallel protein separation before analysis by the mass spectrometer [14,18,19,20] and usually involves using one or multiple types of magnetic nanoparticles with different surface functionalization to selectively adsorb to low-abundance circulating plasma protein without the need for prefractionation or depletion. The proteins bound to the nanoparticles can be digested directly and analyzed by LC-MSMS [21,22,23].

To overcome the limitations and explore the depth of the plasma proteome, we carried out a small-scale plot test using the Proteonano^TM^ (Nanomics Biotech, Hangzhou, Zhejiang, China) beads proteomics workflow [20] and tested a small cohort of plasma samples from PC patients with primary tumors, patients with metastases, and healthy controls. Our findings showed these surface-engineered superparamagnetic nanoparticles can substantially increase the depth of proteome coverage as compared with conventional neat plasma digest strategies. There are large increases in the abundance of ribosomal proteins, amongst other proteins, in the plasma of metastatic cancer patients. Furthermore, a list of 15 potentially cell surface or secreted proteins was identified, and most of these proteins were otherwise undetected in a neat plasma digest. Additionally, we investigated if the same samples analyzed on two different Orbitrap platforms two different column setups—depth and high throughput (HTP)—will produce similar results and/or conclusions. These results offered insights into the application of different instrument platforms and acquisition strategies for discovery and subsequent large-scale analysis of clinical samples.

## 2. Materials and Methods

### 2.1. Patients’ Plasma Collection

The patients’ blood samples were collected in EDTA Vacutainer tubes under the ethics application (No. H2013/04953) approved by the Human Research Ethics Committee at Austin Health. The samples were spun at 400× *g* at 4 °C for 10 min. The resultant supernatants were plasma, which were carefully removed and aliquoted at 500 µL/tube into separate cryovials and stored at −80 °C.

### 2.2. Nanoparticle-Based Enrichment of Plasma and Tryptic Digestion

Sample preparation was carried out according to the manufacturer’s instruction (https://www.nanomics.ai/plasma-kit; accessed on 19 November 2025), except that sample preparation was carried out by hand instead of on a liquid handling system. Briefly, an equal volume of enrichment buffer was added to 40 µL of the plasma samples. The multivalent, multiaffinity nanoprobes (MMNPs) were then added to the plasma mixture and incubated at room temperature (with shaking) for 60 min. Magnetic separation of the MMNPs was then carried out with 2 washes of enrichment buffer. Digestion of bound plasma proteins was carried out directly on beads by adding 20 µL of digestion solution (5 mM DTT in 50 mM NH_4_CO_3_) for 55 min before alkylation with 20 mM iodoacetamide and 2 h digestion with 20 µL (50 ng/µL) sequencing grade modified trypsin (Thermo Pierce, Waltham, MA, USA). The digest was stopped by acidification with 20 µL of 5% Trifluoreacetic acid (TFA) before solid-phased extraction cleanup with C18 stage tips (Affinisep, Le Houlme, France). The cleaned peptides were then freeze-dried and resuspended in 2% acetonitrile containing 0.05% trifluoroacetic acid. Nanodrop peptide estimation of peptides was carried out using the *ε*_205_ = 31 method [24] before analysis by LC MS/MS. Neat plasma was digested using urea-based denaturation followed by C18 stage tip cleanup [25] or the S-Trap protein digestion protocol following manufacturers instructions (https://files.protifi.com/protocols/s-trap-micro-long-4-7.pdf, accessed on 19 November 2025).

### 2.3. LC MS/MS

LC MS/MS analysis was performed either on an Orbitrap Astral or Ascend Mass Spectrometer (ThermoFisher Scientific, Waltham, MA, USA). For the Orbitrap Ascend, the LC system (Ultimate 3000) was equipped with an Acclaim Pepmap nano-trap column (ThermoFisher Scientific, USA) (C18, 100 Å, 75 µm × 2 cm) and an Acclaim Pepmap RSLC analytical column (C18, 100 Å, 75 µm × 50 cm). The tryptic peptides were injected into the enrichment column at an isocratic flow of 5 μL/min of 2% (*v*/*v*) CH_3_CN containing 0.05% (*v*/*v*) TFA for 6 min, applied before the enrichment column was switched in-line with the analytical column. The eluents were 0.1% (*v*/*v*) formic acid (FA) (solvent A) in water and 100% (*v*/*v*) CH_3_CN in 0.1% (*v*/*v*) FA (solvent B), both supplemented with 5% DMSO. The gradient was at 300 nL/min from (i) 0–6 min, 3% B; (ii) 6–7 min, 3–4% B; (iii) 7–82 min, 4–25% B; (iv) 82–86 min, 25–40% B; (v) 86–87 min, 40–80% B; (vi) 87–90 min, 80–3% B; (vii) 90–90.1 min, 80–3% B and equilibrated at 3% B for 10 min before injecting the next sample. For Data Independent Acquisition (DIA) experiments, full MS resolutions were set to 120,000 at *m*/*z* 200 and scanning from 350 to 1400 *m*/*z* in the profile mode. The full MS AGC target was 250% with an IT of 50 ms. The AGC target value for the fragment spectrum was set at 2000%. Fifty windows of 13.7 Da were used with an overlap of 1 Da. Resolution was set to 30,000 and maximum IT to 55 ms. Normalized collision energy was set at 30%. All data were acquired in centroid mode using positive polarity.

For the Orbitrap Astral, the LC system was a Vanquish Neo UHPLC (Thermo Scientific, Waltham, MA, USA) using the heated trap and elute setup. The trap column was an Acclaim Pepmap nano-trap column (Dionex—C18, 100 Å, 75 µm × 2 cm). The analytical column was either a 5.5 cm high-throughput (HTP) µPAC Neo analytical column (Thermo Scientific) or a 50 cm µPAC (depth) Neo analytical column. The eluents were water with 0.1% *v*/*v* FA (solvent A) and 80% CH_3_CN with 0.1% *v*/*v* FA (solvent B). For the high-throughput method using the 5.5 cm µPAC column, the flow rate was 750 nL/min and the gradient was as follows: (i) 0–0.3 min, 3–6% B; (ii) 0.3–23 min, 6–23.5% B; (iii) 23–26.7 min, 23.5–40% B; (iv) 26.7–28.7 min, 40–50% B; and (v) 28.7–28.8 min, 50–99% B. Column wash was then activated at 2 µL/min (combine flow and pressure control) for 1.2 min, followed by trap column wash (combine flow and pressure control with 2 zebra wash cycles at 80% CH_3_CN) and fast equilibration of the analytical column (combine flow and pressure control). For the depth method using the 50 cm µPAC column, the flow rate was 300 nL/min, and the gradients were as follows: (i) 0–46 min, 1.2–28% B; (ii) 46–52 min, 28–50% B; and (iii) 52–57 min, 50–99% B. Column wash was then activated at 0.8 µL/min (combined flow and pressure control) for 3.1 min, followed by trap column wash (combined flow and pressure control with 2 zebra wash cycles at 80% CH_3_CN) and fast equilibration of the analytical column (combined flow and pressure control).

For DIA experiments on the high-throughput setup, full MS resolutions were set to 120,000 at *m*/*z* 200 and scanning from 380 to 980 *m*/*z* in the profile mode. The full MS AGC target was 500% with a maximum IT of 5 ms. DIA was carried out in the Astral analyzer with an isolation window of 2 *m*/*z*, normalized High Energy Collision (HCD) collision energy of 27, normalized AGC target of 500%, and maximum injection time of 3 ms. The cycle time was kept to 0.6 s.

For DIA experiments on the depth setup, full MS resolutions were set to 240,000 at *m*/*z* 200 and scanning from 380 to 980 *m*/*z* in the profile mode. Full MS Automated Gain Control (AGC) target was 500% with a maximum IT of 50 ms. DIA was carried out in the Astral analyzer with an isolation window of 2 *m*/*z*, normalized HCD energy of 25, normalized AGC target of 500%, and maximum injection time of 3.5 ms. The cycle time was kept to 0.6 s.

### 2.4. LC MS/MS

Data Independent Acquisition (DIA) data were analyzed using the direct DIA analysis workflow with default settings on the Spectronaut software (v.19.9.250324.62635) or DIA-NN (v1.8.1) and against the reviewed Uniport *Homo sapiens* database (downloaded October 2024). For Spectronaut searches, trypsin specificity was set to two missed cleavages. Carbamidomethyl (Cys) was defined as a fixed modification, and acetylation (protein N-term) and oxidation (Met) were identified as variable modifications. Results are filtered at a protein and PSM false discovery rate of 1%. Precursor filtering using the Q-value, quantification carried out on the MS2 level, and cross-run normalization strategy set to automatic. For DIA-NN [26] searches, the library-free mode was used. DIA-NN search parameters were as follows: protease, trypsin/P; missed cleavages, 1; peptide length range, 7–30; fragment ion *m*/*z* range, 300–1800; mass accuracy, 0 ppm; modification, cysteine carbamidomethylation, N-term M excision; use isotopologues; heuristic protein inference; and smart profiling library generation. The mass spectrometry proteomics data (identifier PDX065103) are available via the PRIDE archive of the ProteomeXchange Consortium [27].

### 2.5. Informatics

Statistical analysis was carried out using Perseus version 2.0.7.0 [28]. Differentially expressed proteins were identified using a two-sample Student’s *t*-test (*p* < 0.01) or in combination with a permutation-based correction for multiple hypothesis testing with false discovery rate, FDR = 0.05 and S0 = 1. STRING [29] analysis was carried out using Markov clustering (MCL, inflation parameter of 3) and a minimum required interaction score of 0.7 (high confidence). A clustering map was generated using Euclidean distance and the complete linkage clustering method with data from all significant proteins (metastasized and healthy control). Each row in the color heatmap indicates a single protein based on the Human UniProt accession number. Normalized protein abundance values (Z-scores) are indicated colorimetrically for each protein as the deviation from the mean by standard deviation units. A priori power analysis indicates that with *n* = 4 per group, is sufficient to detect an effect size of >2-fold differences with 80% power, α = 0.05 (CV 50%).

## 3. Results

### 3.1. Identification of Proteins from Nanoparticle Enrichment and Neat Plasma

We performed a small-scale pilot test using nanoparticle-based enrichment of plasma proteins from a total of 15 clinical samples. These samples consisted of four healthy controls, six pancreatic cancer patients with primary tumors, and five metastatic pancreatic cancer patients. Within the metastatic group, four had metastasized to the liver and one had metastasized to the peritoneal omentum (Table 1). An equal amount of Proteonano^TM^ (Nanomics Biotech, Zhejiang, China) magnetic nanoparticles were added to these plasma samples, and digested plasma samples were analyzed in a data-independent (DIA) workflow comprising two different Orbitrap mass spectrometers and three different analytical columns (Figure 1). On the Orbitrap Ascend, we used the standard DIA pipeline in the laboratory, focusing on depth of identification (97 min injection to injection, 14 Samples Per Day [SPD]). Two different acquisition strategies were evaluated on the Orbitrap Astral—depth (72 min injection to injection, 20 SPD) or high throughput, HTP (38 min injection to injection, 37 SPD), to increase throughput (Figure 1).

Using the Orbitrap Astral, 5934 proteins were collectively identified using the Spectronaut search engine across all 15 samples using the depth strategy (Figure 2A, Table 2). There were 4940 proteins identified on the Orbitrap Ascend and 3734 using the HTP method on the Orbitrap Astral. As deconvolution of multiplexed fragment ion spectra to connect the precursor and fragment ions is important and performed differently by different search engines [30], we performed the same database search using a separate search engine [26] to compare and contrast the rates of protein identification. This is not an exercise to define which search engine gave the best data but to demonstrate the difference in the number of identified proteins using different search engines. When using DIA-NN, the Orbitrap Astral data identified 6958 proteins and 4684 proteins across all 15 samples using the depth and HTP strategies, respectively (Table 2). Using the DIA-NN search engine, there were 4794 proteins identified on the Orbitrap Ascend.

To further increase the identification rate, we pooled all 15 samples and performed orthogonal basic reversed-phase (bRP) fractionation into eight different fractions. The number of protein identifications on both the ‘depth’ acquisition methodology on Orbitrap Astral or Ascend were similar compared to direct analysis of the unfractionated samples, suggesting we may be approaching the limit of detection for all proteins enriched by magnetic nanoparticles (Table 2). In contrast, the bRP fractionated samples on the Orbitrap Astral HTP acquisition strategy increased protein identifications by ~30%, suggesting that the substantially greater sample complexity of unfractionated samples reduced the capacity of the workflow to identify proteins in this workflow.

We then investigated the differences between magnetic nanoparticles enriched control plasma and that of S-trap digested neat control plasma (Figure 2B). There are more than 7× more proteins that were identified from the magnetic nanoparticle-enriched samples. Even though 79 proteins appeared to be identified solely from the neat control plasma samples, a thorough investigation revealed all but 1 were protein isoforms and were not picked up using exact keyword searches. We then compared the 5934 and 6958 proteins that were identified using the depth strategy on Orbitrap Astral (from 2 different search engines) against the 2021 build of 5092 proteins from the Human Plasma Proteome Project (HPPP) [31] and the list of 4608 canonical human plasma proteins downloaded from the Human Plasma PeptideAtlas (https://db.systemsbiology.net/sbeams/cgi/PeptideAtlas/GetProteins?atlas_build_id=559&organism_id=2&redundancy_constraint=4&presence_level_constraint=1&action=QUERYPMID: 34672606, accessed 19 November 2025). Of these 2 builds, 3241 proteins were present in all 4 datasets (Figure 2C). There were approximately 1800 proteins solely identified from the nanoparticle-enriched samples. All proteins identified from the nanoparticle-enriched normal plasma were then mapped to the HPPP abundance data, which were calculated based on the spectral counting of the total number of PSM events associated with each protein (Appendix A) and high dynamic range from the very high to low abundance proteins. The neat, digested plasma samples, in contrast, had a significantly reduced proportion of lower abundance proteins.

### 3.2. Comparing the Plasma Proteome of Individuals with Primary or Metastatic PC and Healthy Controls, Acquired Using Orbitrap Astral (Depth)

Despite the higher number of protein identifications using DIA-NN, subsequent analyses were all carried out using Spectronaut, as our laboratory already has a robust workflow using this search engine. We focused on data acquired using the Orbitrap Astral (depth) and carried out statistical analyses to compare plasma samples from individuals with different PC stages with those of healthy controls. We restricted the analysis to only including proteins that were detected and quantified in all 15 samples to ensure comparisons were only for proteins that were consistently identified in all samples, rather than the effect of differential enrichment, sample collection, or processes. However, given the small sample size, these findings should be considered preliminary. Proteins were deemed significantly changed based on a two-sample s0 modified Student’s *t*-test (s0 = 1) [28] in combination with a permutation-based correction for multiple hypothesis testing (FDR = 0.05). When comparing the plasma proteome of PC patients that had metastasis to that of control plasma, we observed a statistically significant difference in 146 proteins that were higher in individuals with PC who had metastasis (Figure 3A,B). Of these 146 proteins, only 9 were identified in the neat, digested control plasma samples. The top 10 highest changed proteins range from 29 to 77 times higher in metastasis patients compared to healthy controls. STRING MCL [29] clustering network of the 146 significantly changed proteins revealed four distinct clusters: peptide chain elongation, mRNA processing, structural constituents of chromatins, and fatty acid degradation (Figure 4). Gene ontology showed the most significant enrichment being the structural constituent of ribosomes. Comparing the plasma proteome of PC patients with the primary tumor to the healthy controls, 82 proteins were deemed significantly changed in abundance (Appendix A). Of these 82 proteins, only 4 were identified in the neat digested control plasmas, and, similar to the metastasis patients, both ADH1C and ADH1B were also the most changed proteins at 28 and 20 times, respectively. STRING MCL of these 82 proteins showed clustering to mitochondrial ATP synthesis and apoptosis-induced DNA fragmentation (Appendix A). The difference between the plasma proteome of metastasis and primary tumor is less profound, with 25 proteins deemed significantly changed based on two-sample Student’s *t*-test (*p* < 0.01) (Appendix A). STRING MCL of these 25 proteins showed clustering to cytoplasmic translation and structural constituents of chromatins (Appendix A).

### 3.3. Comparing the Plasma Proteome of Individuals with Primary or Metastatic PC and Healthy Controls, Acquired Using Orbitrap Ascend and Orbitrap Astral (HTP)

We next acquired the same 15 clinical samples in parallel on the Orbitrap Ascend (depth method, 50 cm PepMap column) and Orbitrap Astral (HTP method, 5.5 cm µPAC column). We then assessed whether the differentially regulated proteins identified in these setups would lead to a similar biological conclusion as compared to that of the Orbitrap Astral (depth). On the Orbitrap Ascend, a total of 4940 proteins were identified, and on the Orbitrap Astral HTP method, a total of 3734 proteins were identified (Figure 2A, Table 2).

Comparing the plasma proteome of individuals with metastatic PC with healthy controls on the Orbitrap Ascend, a total of 95 proteins were significantly changed, based on the s0-modified (s0 = 1) Student’s *t*-test in combination with a permutation-based correction for multiple hypothesis testing (FDR = 0.05) (Figure 5A). In contrast, there were 33 significantly changed proteins identified using the HTP acquisition methodology on the Orbitrap Astral. Focusing only on the metastatic PC samples vs. healthy controls, the two lists of significantly changed proteins were subjected to STRING protein-protein association network and functional enrichment analysis. From the 95 proteins that were deemed significant on the Orbitrap Ascend, the top 3 clusters from the MCL analysis were peptide chain elongation, spliceosome, and structural constituent of chromatins (Figure 5B). Cytoplasmic translation was the most significant Gene Ontology enrichment. From the 33 proteins that were deemed significant on the Orbitrap Astral (HTP), the top 3 clusters from the MCL analysis were DNA damage, peptide chain elongation, and mRNA processing (Figure 5C). Nucleosome assembly was the only significant Gene Ontology enrichment.

## 4. Discussion

Nanoparticle-based plasma proteome profiling has recently gained popularity and increased in applications [18,19,20,32,33,34]. When in combination with sample prefractionation strategies and newer generation mass spectrometers, the number of plasma proteins identified has increased from a few hundred to more than 10,000 [34]. These developments have renewed interest in plasma as a matrix to detect circulating proteins for the discovery of biomarkers for disease diagnosis or simply understanding the functions of proteins in blood. In this pilot small-scale trial, we obtained plasma samples from three groups: healthy controls and individuals with primary and metastatic PC, and we evaluated if there were any differences in the plasma proteome profiles between these three groups. We further evaluated the differences in using different MS acquisition strategies (depth or HTP) on the Orbitrap Astral and evaluating the same sample analyzed on an earlier version of the Orbitrap mass spectrometer (Ascend).

We first assessed the proteome coverage afforded by these magnetic nanoparticles on the Orbitrap Astral using the depth strategy. A total of 5934 proteins were identified using the Spectronaut search engine in contrast to just 697 proteins identified from the neat plasma digest on the same instrument platform. These results agree with earlier plasma nanoparticle-based enrichment publications [18,19,32,33]. Processing these same data using DIA-NN gave ~17% more protein identification. Various DIA search engines have been extensively compared [30], and a thorough treatment of the reasons for these differences is beyond the scope of this work.

We then conducted basic reversed-phase fractionation of the pooled samples into 8 different fractions and processed them on the same 3 MS acquisition strategies (Figure 1). We observed a modest (~3–6%) increase in protein identifications for samples acquired on the Orbitrap Ascend or Orbitrap Astral (depth) (Table 2). This could suggest we are at or close to the limit of detection of proteins enriched from these magnetic nanoparticles. Narrow window (2 Da) isolation strategies, which have been reported to exhibit superior performance in library-free searches, could also play a role [30,35]. The increase for Orbitrap Astral (HTP), in contrast, was more substantial (31%), suggesting lower resolution of the HTP methodology possibly affecting proteome depth when coupled with fast data acquisition.

The Orbitrap Astral (depth) data searched with the Spectronaut and DIA-NN software were then compared against the latest Human Plasma PeptideAtlas [36] and Human Plasma Proteome Project (HPPP) 2021 build, which is a cumulative sum of plasma proteins identified from 240 MS-based experiments [31]. We identified 3241 proteins in common across all 4 datasets derived from nanoparticle enrichment (Figure 2C), and each has a different set of proteins that are uniquely identified. More than 300 of these identified proteins are on the list of FDA-approved drug targets, compiled in the Human Protein Atlas [37]. In contrast, only 88 proteins detected in the neat plasma digest are approved drug targets, indicating the advantage of significantly increased depth of coverage achievable using nanoparticle enrichment strategies. It should be noted that differences in sample preparation (enrichment, depletion, fractionation, etc.) and acquisition methodologies can have a profound impact on the results. Two recent publications [34,38] have shown that peripheral blood mononuclear cells (PBMC), platelet, or erythrocyte contamination in magnetic nanoparticle enrichment experiments could lead to very high reported protein identifications, even at low levels. Magnetic-based enrichment methodology (Mag-Net) also can capture extracellular vesicles (EV) from plasma [39,40] which will also impact the number of protein identifications. Conversely, the presence of these plasma ‘contaminants’ has been suggested to be markers of immune escape or thrombocythemia from early stages of cancer [41,42].

Of the 146 significantly changed proteins (Spectronaut dataset) that are more abundant (3- to 77-fold) in individuals with metastatic PC compared to that of healthy controls, only 9 were detected in the neat digested plasma samples. A total of 89% of these proteins were in both the Human Plasma Peptide Atlas and HPPP database, and 65% of these proteins were on the Protein Atlas plasma protein database. Eight of these proteins (IDH1, ADH1B, ADH1C, SHBG, EEF2, PKLR, CPS1, and RPL13A) are already on the list of FDA-approved drug targets, compiled in the Human Protein Atlas, with information from the DrugBank [37,43]. Interestingly, three of these eight proteins, ADH1C, ADH1B, and CPS1, are among the top 10 most differentially regulated proteins, at 77, 45, and 44 times higher than healthy controls, respectively. Alcohol dehydrogenase, ADH1C and ADH1B, are members of the alcohol dehydrogenase family and are associated with many cancers, especially hepatocellular carcinomas [44,45]. ADH1C is involved in tumor immune cell infiltration [46], and ADH1B is associated with increased risk of breast and gastric cancer [47,48]. These proteins are the most differentially regulated and were identified in four out of five individuals who had PC liver metastasis. Carbamoyl phosphate synthase 1 (CPS1) converts ammonia into carbamoyl phosphate, and the expression of this urea cycle enzyme is altered in cancers and is associated with prognosis [49]. Knocking down of CPS1 in combination with chemotherapy reduces cell viability in vitro, and small-molecule CPS1 inhibitors have been developed [50].

Blood diagnostic tests target proteins that are secreted by cells into plasma as a response to various stimuli. The biological and clinical behavior of primary and metastatic cancer differs from person to person. The ability to detect these circulating protein biomarkers is essential to help clinicians choose proper treatments for patients [51]. One useful property of circulating proteins is their ability to act as sensor molecules, including proteins, cytokines, growth factors, extracellular matrix, exosomes, etc., that are secreted or released by cancer cells. We cross-referenced the 146 significantly changed proteins that were higher in individuals with metastatic PC vs. that of healthy controls with the in silico human Surfaceome [52] and Human Protein Atlas blood Secretome database [37] and narrowed down to 15 secreted or cell surface proteins (Table 3). In light of the small sample size, these findings should be viewed as tentative and interpreted with caution. Interestingly, seven of these proteins are known clinical or preclinical biomarkers based on data collated by MarkerDB [53] and the National Institute of Health (NIH)—early detection research network (NIH-EDRN) [2]. One protein (NT5E) is in a stage one clinical trial, three of these proteins (LRG1, MDK, and RETN) are in stage 2 clinical trials, and two of these proteins (IGFBP2 and PRL) are in stage 3 clinical trials. Excluding the 7 secreted proteins, there are another 15 proteins from the list of 146 proteins that have been cataloged in MarkerDB and the NIH-EDRN databases in addition to being in the Human Plasma Peptide Atlas and/or HPPP database, making them valuable targets for future validations. Importantly, 10 out of these 15 secreted or cell surface proteins were not identified in neat plasma, demonstrating the importance of the depth of proteome afforded by the nanoparticle enrichment methodology. Future validation experiments on a separate clinical cohort using targeted Single Ion Monitoring (SRM) mass spectrometry assays with synthetic heavy-labeled peptides [54] and antibody-based assays (Western blot and ELISA) will be carried out to validate the abundance level of these proteins before large-scale screening assays to assess the utility of these surface proteins.

When all 146 significantly changed proteins were subjected to an unbiased STRING protein-protein network analysis [29], which was constructed using only a high minimum required interaction score of 0.7 (and hiding disconnected nodes), the top four main gene clusters identified using Markov clustering (MCL) are related to peptide chain elongation, mRNA processing, structural constituents of chromatin, and fatty acid degradation. Gene ontology showed enrichment of structural constituents of ribosomes and proteins involved in RNA (rRNA and mRNA) or nucleic acid binding. The presence of constituents of ribosomes could be due to platelets, PMBCs, or cellular damage or death, where these proteins are leaked into the plasma. However, upon checking the plasma quality panel contamination index [63], comprising 30 proteins each, for potential erythrocytes, platelets, and coagulation, only one protein (SCL4A1) is found. We also do not find any significant changes in known cell lysis protein markers such as lactate dehydrogenase (LDH) in our data. Hyperactivation of ribosome biogenesis is one of the many characteristics of cancers, which is necessary to support protein synthesis due to the unregulated cell growth [64]. Extracellular ribosome fragments have been previously demonstrated transiently in the media of cultured mammalian cells [65]. The presence of non-vesicular extracellular RNA in human plasma has also been reported [66], and all of these suggest unknown secretion mechanisms or pathways associated with cancer metastasis. A potential source of these ribosomal proteins, however, could originate from extracellular vesicles (EVs) [39,67]. Using magnetic-based enrichment methodology, EVs from plasma can be captured through a combination of size and charge strategies. More than 4000 proteins in the Wu et al. [39] study were identified from a cohort of 40 individuals. The EVs were categorized into the following three main groups: exosomes, microvesicles, and cellular debris/apoptotic bodies. There is a total of 13 protein markers for these three main groups, including classical markers like CD9, CD63, FLOT1, FLOT2, TSG101, CD40, ATP5F1A, HSP90B1, and ANXA5. In their Mag-Net-based enrichment, the authors were able to demonstrate a greater than 16–30-fold increase in the abundance of these three groups as compared to neat plasma. All except one protein from the three groups of EVs have been identified in our dataset, and statistical enrichment of the 146 significantly changed proteins using the FUNRICH tool [68] revealed enrichment of protein exosomes (Appendix A), and transmission electron microscopy (TEM) analysis of the nanoparticles for the presence of EVs has been planned to verify this observation. None of the 13 classical protein markers, however, are statistically different in individuals with metastatic PC vs. that of healthy controls.

We also evaluated whether running the same samples on different MS instrumentations could give us comparable conclusions. The Orbitrap Ascend, introduced in 2023, is the latest in the series of Orbitrap Tribrid mass spectrometers [69], possessing significant improvements over earlier generations [70]. With the introduction of the new Orbitrap Astral [21], multiple groups have reported impressive depth using a short analytical time scale of only 7.8 min for HTP studies [21] or significant increased depth of proteome coverage [71,72,73] as compared to the Orbitrap Ascend. Even though the technological advancement from the latest Tribrid instrument is substantial (vs. Orbitrap Eclipse), there are many laboratories, including ourselves, operating 2–3 year old instrumentation. We compared key parameters of our experimental data that would influence the biological interpretation between different instrument parameters. We assessed the overlap of identified proteins, the numbers of differentially regulated proteins, the number of known FDA-validated targets, and whether unbiased clustering using STRING yields similar results. It is apparent that the Astral HTP methodology identified the fewest proteins (Figure 2A). There are fewer significantly changed proteins in the Astral HTP methodology, although it is notable that the statistical test used for this is different from the other two methodologies, as applying permutation-based correction for multiple hypothesis testing yields no hits. Five proteins appear to be significant only in the Astral HTP methodology, although these proteins were identified in the other two methodologies but were not statistically different. The number of proteins identified using the Ascend is lower than the Astral (depth) methodology, but there are still 220 proteins (the number is lower when considering classification of different isoforms of the same proteins) that are uniquely found in the Ascend, likely due to the different analytical columns used and longer HPLC elution time. We then examined the list of 15 cell surface proteins (Table 3), and the Orbitrap Ascend identified 7 and Astral (HTP) identified 3. Because of the smaller number of significantly changed proteins fed into STRING, the results, although not the same, were generally in consensus. All 3 STRING clustering showed enrichment and clustering of proteins involved in peptide chain elongation, mRNA processing (spliceosome), and constituents of chromatin (DNA damage). These data highlight that similar conclusions can potentially be made, although the Astral (depth) methodology does give one more comprehensive data when compared to the older instrumentation or faster acquisition on the same instrument.

## 5. Conclusions

In this small-scale pilot evaluation, we have demonstrated the utility of nanoparticle-based enrichment strategies to significantly increase the depth of the plasma proteome in patients with pancreatic cancer. Thousands of additional proteins that would otherwise be undetectable due to the dynamic range issues were now visible. The identification of previously established clinical and preclinical protein markers in individuals with metastatic pancreatic cancer, where they would otherwise go undetected in neat plasma, is very encouraging. We acknowledge, however, that the relatively small sample size limits the statistical power of our findings, and future studies with larger cohorts will be carried out to address this limitation.

## Figures and Tables

**Figure 1 cancers-17-03765-f001:**
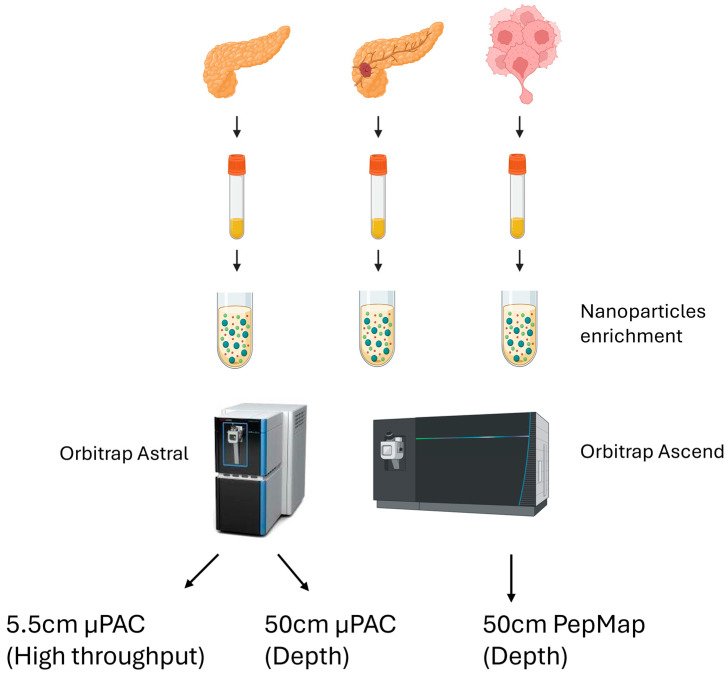
Proteomics sample preparation workflow on all three patient types and the acquisition methodologies, using two different Orbitrap mass spectrometer platforms and column setup.

**Figure 2 cancers-17-03765-f002:**
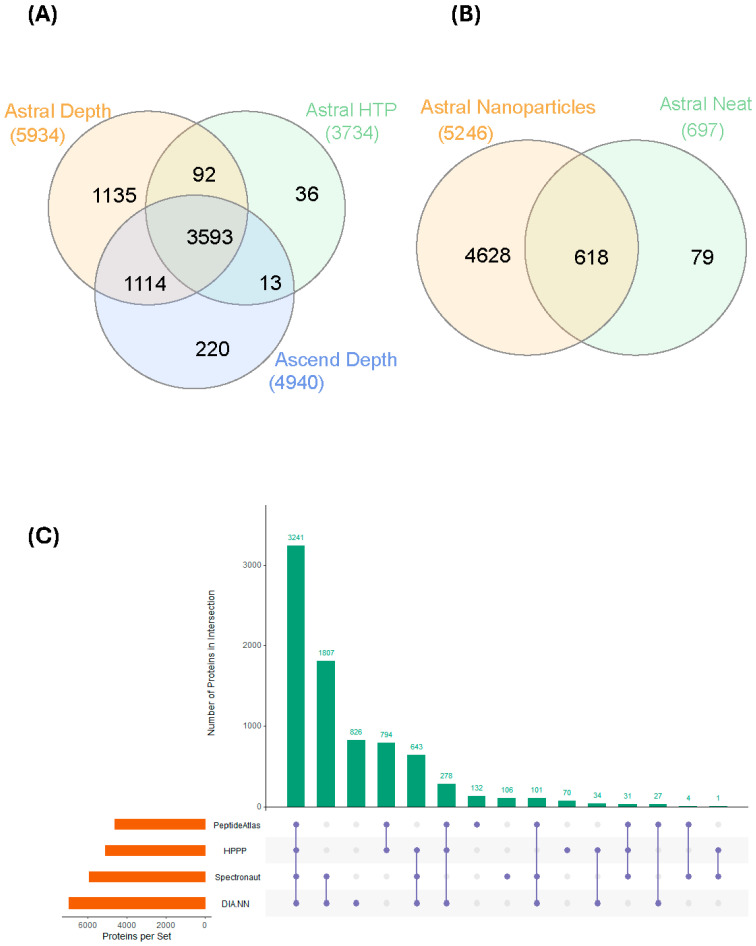
Protein identification based on the different acquisition methodologies. (**A**) Comparing the Depth (Astral, Ascend) and HTP (Astral) strategies; (**B**) comparing number of proteins identified from healthy individuals from magnetic nanoparticle-enriched samples with that from neat plasma digest using the Astral Depth procedure; and (**C**) comparing the number of identified proteins using two different search engines, Spectronaut and DIA-NN, and overlaying the identified proteins with the HPPP and Human Plasma PeptideAtlas databases.

**Figure 3 cancers-17-03765-f003:**
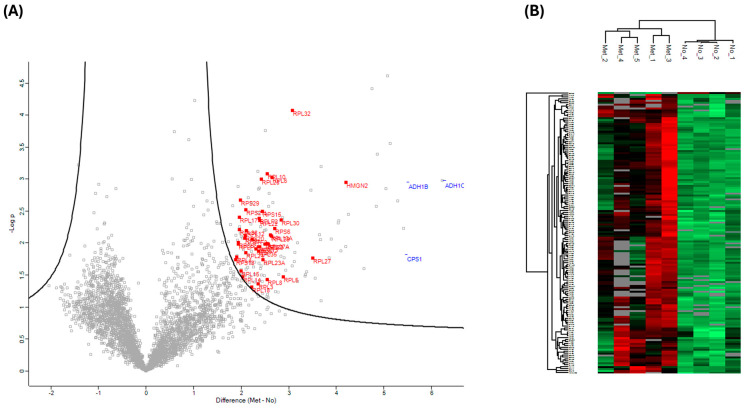
Statistical analysis of individuals with PC that had metastasized vs. that of healthy controls. (**A**) Volcano plots were generated based on two-sample Student’s *t*-tests in combination with a permutation-based correction for multiple hypothesis testing (FDR = 0.05) and an S0 = 1 filter. Proteins outside the solid curved line are deemed to be statistically significant. Ribosomal proteins (red square) are highlighted. (**B**) A clustering map was generated using Euclidean distance and the complete linkage clustering method with data from all significant proteins (metastasized and healthy control). Each row in the color heatmap indicates a single protein based on the Human UniProt accession number. Normalized protein abundance values (Z-scores) are indicated colorimetrically for each protein as the deviation from the mean by standard deviation units.

**Figure 4 cancers-17-03765-f004:**
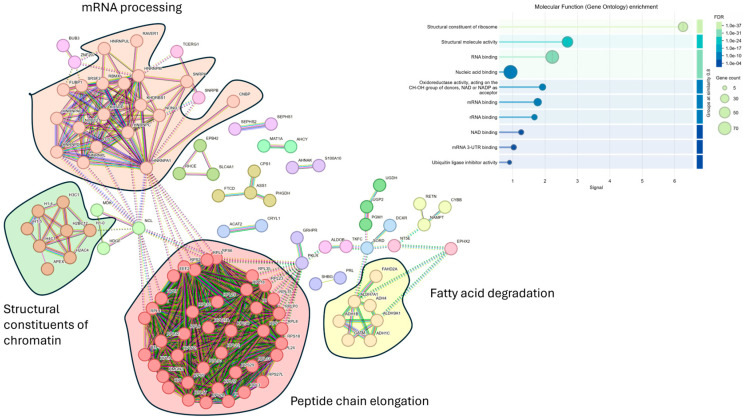
Visualization of known interactions using the STRING database. A total of 146 significantly changed proteins of individuals with PC that had metastasized vs. that of healthy controls were used in the analysis. Four main clusters were revealed from MCL and insert, showing the molecular function (gene ontology) enrichment. PPI enrichment *p*-value = <1.0 × 10^−16^.

**Figure 5 cancers-17-03765-f005:**
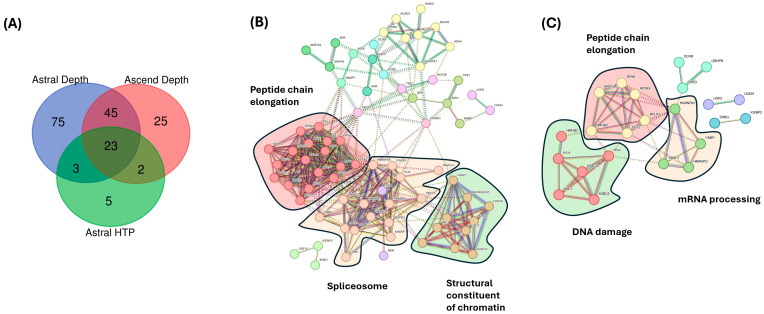
Analysis of significantly changed proteins using different MS acquisition methodologies. (**A**) Comparing the significantly changed proteins of individuals with PC that had metastasized vs. that of healthy controls across three acquisition methodologies (Astral Depth, Astral HTP, and Ascend). (**B**) Three main clusters were revealed from MCL enrichment for significant proteins from Orbitrap Ascend. PPI enrichment *p*-value = <1.0 × 10^−16^. (**C**) Three main clusters were revealed from MCL and insert, showing enrichment for significant proteins from Orbitrap Astral (HTP). PPI enrichment *p*-value = <1.9 × 10^−11^.

**Table 1 cancers-17-03765-t001:** Patient characteristics.

Label	Age	Sex	Organs Cancer Metastasized to	Group
No_1	55	F	None	Non-PC samples
No_2	33	F	None	
No_3	42	M	None	
No_4	31	M	None	
Pri_1	70	F	None	PC (primary tumor)
Pri_2	60	M	None	
Pri_3	69	M	None	
Pri_4	80	F	None	
Pri_5	66	M	None	
Pri_6	79	M	None	
Met_1	34	M	Liver metastases	PC (metastases)
Met_2	74	F	Liver metastases	
Met_3	69	F	Liver metastases	
Met_4	66	M	Liver metastases	
Met_5	70	F	Peritoneal metastases	

**Table 2 cancers-17-03765-t002:** Comparing the number of identified proteins across three different acquisition methodologies using two different search engines and basic reversed (bRP) fractionation of the pooled samples.

Spectronaut (Sparse Prot ID)	Ascend	Astral (HTP)	Astral (Depth)
Direct DIA (15 samples)	4940	3734	5934
8 pooled bRP fractions	5248	4896	6166
Neat Plasma	775	387	697
8 pooled Neat Plasma bRP fractions	1437	660	1267
DIA-NN	Ascend	Astral (HTP)	Astral (Depth)
Direct DIA (15 samples)	4794	4684	6957
8 pooled bRP fractions	4695	5369	6401
Neat Plasma	607	352	742
8 pooled Neat Plasma bRP fractions	1050	626	1052

**Table 3 cancers-17-03765-t003:** List of 15 predicted or experimentally determined secreted or cell surface proteins that are significantly changed in individuals with PC that had metastasis vs. that of healthy controls.

Gene	UniProt	Protein Description	Localization ^	Met vs. Control ^#^	MarkerDB ^$^	NIH-EDRN *	ID ^@^	Comment
CCDC126	Q96EE4	Coiled-coil domain-containing protein 126	SEC	4.6			AD	Potential predictor pancreatic cancer [55]
CYBB	P04839	NADPH oxidase 2	SUR	7.2			AD	Potential prognostic target in glioma [56]
FGL1	Q08830	Fibrinogen-like protein 1	SEC	6.6			AD	Potential marker for diagnostic and prognosis [57]
GYPC	P04921	Glycophorin-C	SUR	4.0			AD	Potential marker for breast and ovarian cancer (PMID: 39497123)
HDGF	P51858	Hepatoma-derived growth factor	SEC	5.8			AD	Potential marker for broad range cancer [58]
IGFBP2	P18065	Insulin-like growth factor-binding protein 2	SEC	6.9		2, 3	AD AH AS	Breast, Colon, Lung, Ovary (NIH-EDRN data)
LRG1	P02750	Leucine-rich alpha-2-glycoprotein	SUR/SEC	5.9		1, 2	AD AS	Breast, Ovary, Pancreas (NIH-EDRN data)
MDK	P21741	Midkine	SEC	10.4		2	AD AH AS	Potential diagnostic and prognostic cancer marker [59]
NAMPT	P43490	Nicotinamide phosphoribosyltransferase	SEC	5.4			AD AS	Over expression in cancer and potential target [60]
NT5E	P21589	5′-nucleotidase	SUR	5.5		1	AD AS	Prostate (NIH-EDRN data). Potential prognostic in pan-cancer [61]
PRL	P01236	Prolactin	SEC	12.7	Yes	3	AD	Mayo Clinic. Breast, Ovary, Pancreas (NIH-EDRN data)
RETN	Q9HD89	Resistin	SEC	4.5		2	AD	Breast (NIH-EDRN data)
SHBG	P04278	Sex hormone-binding globulin	SEC	6.2	Yes		AD AH AS	Mayo Clinic
SLC4A1	P02730	Band 3 anion transport protein	SUR	4.0			AD AS	Correlate with gastric cancer progression [62]
TSPAN18	Q96SJ8	Tetraspanin-18	SUR	−2.7			AD	

^ Localization (SEC—secreted, SUR—surface). ^#^ Fold change (Metastasized PC vs. Normal Control). ^$^ Proteins in the MarkersDB database. * Clinical trial stage(s) as reported in the NIH Early Detection Research Network (NIH-EDRN). ^@^ AD (Astral Depth), AH (Astral HTP), AS (Ascend).

## Data Availability

The mass spectrometry proteomics data (identifier PDX065103) are available via the PRIDE archive of the ProteomeXchange Consortium [27].

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
