# Peer review of "Evaluation of Nanoparticle-Based Plasma Enrichment on Individuals with Primary and Metastatic Pancreatic Cancer"

_cancers, 2025, doi:10.3390/cancers17233765_

Round 1
Reviewer 1 Report (Previous Reviewer 1)
Comments and Suggestions for Authors
All my previous requests have been addressed except for experimental validation, which the authors now clearly acknowledge as a limitation.
As a pilot study, this is acceptable.
The manuscript is clear, balanced, and technically sound, and can be accepted for publication.
Author Response
We sincerely thank you for going through the revised manuscript and the positive feedback.
Reviewer 2 Report (Previous Reviewer 2)
Comments and Suggestions for Authors
The manuscript is not about technology or methodology. A commercial kit with nanobodies is used to enrich informative proteins, which is the whole purpose of these commercial nanobodies. The protein isolates are then analysed on different commercial mass spectrometers. So, where is the news?
No repetitions were done of the protein isolation so as to validate reproducibility of the process. Would always be the same proteins been isolated and would their ratio to each other remain the same between repetitive isolations from the same plasma samples?
Does the enrichment kit affect protein abundances? By comparison of the “neat control plasma”, this could be studied for the several hundred proteins that are identified by both the “neat” and the enriched isolates. By the way, there is no protocol of how the “neat” isolates were done.
The manuscript is also not about marker identification. For a start, the sample number is totally insufficient. Individual rather than disease-related variations between patients cannot be excluded this way. Also, the question of reproducibility is unanswered.
Last, any potential markers identified in a discovery group of plasma samples has to be validated in a separate, independent group of plasma samples. Otherwise, the conclusion is not any more than speculation.
In conclusion, the paper lacks novelty in every aspect. Also, the reported results are very unlikely to be robust and accurate in terms of technical reproducibility, let alone biomedical relevance.
Author Response
Thank you for your time and constructive comments on our revised manuscript. We have carefully thought through your comments and provided a through answer below.
1) We fully understand the concerns of reviewer 2 on the needs to have larger sample size and more validations to identify the biomarker.
Plasma is among the most challenging biological matrices to analyze using proteomics. Historically, neat plasma proteomics - whether for pancreatic cancer or other malignancies - have plateaued at only a few hundred identified proteins because of its vast dynamic range and the dominance of high-abundance proteins, severely limiting its diagnostic potential. Even a very recent publication using modern instrumentation and high-abundance protein depletion (Tognetti et al., 2025, iScience 28:4) quantified only 1,011 proteins across 211 samples. Although our samples are not directly comparable, our workflow enabled the identification of > 5,000 proteins from neat plasma.
The Proteonano methodology we evaluated is cost-effective and accessible to most laboratories, providing a practical route to deep plasma proteomics without the need for costly depletion or extensive fractionation. As noted by Reviewer 3, there remains an urgent need for more sensitive analytical techniques to overcome the inherent challenges of plasma’s wide dynamic range—our pilot study directly addresses this gap.
“Reviewer 3: The topic is timely, relevant, and scientifically significant, especially given the urgent need for more sensitive and specific plasma biomarkers in pancreatic cancer.
We have explained in detail that this is a pilot study to demonstrate the potential of this new methodology in identifying proteins that were previously undetected in a traditional ‘single shot’ plasma proteomics. We understand the need for larger sample size but is something we are unable to do right now due to the availability of suitable donors and cost involved. On the advice of reviewers 1 and 3, we have in the manuscript highlighted the limitations, planned validation future experiment and have also included power analysis for cohort of this size (line 180).
“Reviewer 1: As a pilot study, this is acceptable. The manuscript is clear, balanced, and technically sound, and can be accepted for publication”
“Reviewer 3 Despite the clear strengths, the study is limited by a small sample size and lacks independent validation, which restricts the generalizability of the conclusions. However, the technical rigor and innovation make this a valuable contribution to the proteomics and biomarker discovery community”.
“The pilot study includes only 15 samples (4 controls, 6 primary, 5 metastatic), which severely limits statistical significance. The authors correctly acknowledge this but should include a power analysis or justification for cohort size in the Methods or Discussion.”
2) In term of this is just another commercial kit to enrich for informative proteins,
“A commercial kit with nanobodies is used to enrich informative proteins, which is the whole purpose of these commercial nanobodies. The protein isolates are then analysed on different commercial mass spectrometers. So, where is the news”
We believe Reviewer 2 may have misunderstood the technology used in our study. The nanoparticle-based enrichment methodology applied here is fundamentally different from antibody or aptamer based commercial assays such as Olink or SomaScan. Those platforms rely on affinity reagents (antibodies or aptamers) to selectively capture a predefined list of plasma proteins. While powerful, they are limited by binding specificity, dynamic range, coverage, and cost, and are typically restricted to thousands of pre-selected targets.
In contrast, our study uses magnetic nanoparticles with tailored surface functional groups designed to selectively adsorb low abundance circulating plasma proteins through physicochemical (non-covalent) interactions, followed by identification by mass spectrometry. The workflow is fully compatible with automation, requires no high-abundance protein depletion or complex fractionation, and is therefore lower in cost, higher in reproducibility, and easier to implement than antibody/aptamer-based assay.
We like to emphasize that we have no financial or proprietary interest in any material discussed in this article. We have adopted this approach for plasma proteomics in pancreatic cancer research precisely because the entry cost for affinity-based platforms (e.g., Olink or SomaScan, often >USD$1,000 per sample in Australia) remains prohibitive for most laboratories. Our workflow provides a cost-effective alternative that can be adopted widely without specialized reagents.
3) On issue of no repetitions were done of the protein isolation so as to validate reproducibility of the process.
We have performed stringent and robust statistical analysis on the biological replicates using S0-modified (S0=1) two-sample Student’s t-test (Tusher and Chu PNAS 2001) combined with a permutation-based correction for multiple hypothesis testing at a false discovery rate (FDR) of 0.05. This workflow, implemented in Perseus (Tyanova et al., Nat Proc 2016), includes a randomization procedure in which group labels are shuffled to generate a null distribution, enabling robust FDR control and minimizing false positives. This approach ensures that detected protein changes reflect protein changes within these 15 samples reflect genuine relative differences and has been widely applied in high-impact comparative proteomics studies where replicate size of 3 or 4 per group is common. Additionally, we have now included an a priori power analysis in the methods (line 180) on the reliability to detect an effect of at least 2-fold change (80% power, α = 0.05) based on the sample size of the smallest sample group of n=4 (control). The neat plasma digestion protocol using the S-trap methodology is now referenced.
Reviewer 3 Report (New Reviewer)
Comments and Suggestions for Authors
The manuscript presents a well-structured and methodologically detailed study evaluating the use of nanoparticle-based enrichment (Proteonano™) to enhance plasma proteome coverage in pancreatic cancer (PC) patients. The study’s findings demonstrate a remarkable increase in protein identification, particularly of cancer-associated and secreted proteins, when compared to neat plasma digests. The topic is timely, relevant, and scientifically significant, especially given the urgent need for more sensitive and specific plasma biomarkers in pancreatic cancer.
Despite the clear strengths, the study is limited by a small sample size and lacks independent validation, which restricts the generalizability of the conclusions. However, the technical rigor and innovation make this a valuable contribution to the proteomics and biomarker discovery community.
My specific comments for further improvement:
(1) The pilot study includes only 15 samples (4 controls, 6 primary, 5 metastatic), which severely limits statistical significance. The authors correctly acknowledge this but should include a power analysis or justification for cohort size in the Methods or Discussion.
(2) Validation of candidate biomarkers (e.g., ADH1C, LRG1, MDK) via orthogonal methods (e.g., ELISA or targeted MS) is missing. At least a brief outline of planned validation experiments should be added to strengthen translational impact.
(3) The authors propose that extracellular ribosomal proteins could be due to EVs or cancer-related secretion. However, this conclusion remains speculative.
(4) Lines 55–57: Add a reference or sentence quantifying the “large dynamic range” of plasma proteins.
(5) Table 3: Add Uniprot IDs or accession numbers for completeness.
(6) Abbreviations: Ensure all are defined upon first use (e.g., DIA, SPD).
(7) Data availability statement: Confirm that PRIDE accession PDX065103 is active before publication.
Author Response
Thank you for your positive feedback and constructive comments. We have addressed your concerns point by point below.
The manuscript presents a well-structured and methodologically detailed study evaluating the use of nanoparticle-based enrichment (Proteonano™) to enhance plasma proteome coverage in pancreatic cancer (PC) patients. The study’s findings demonstrate a remarkable increase in protein identification, particularly of cancer-associated and secreted proteins, when compared to neat plasma digests. The topic is timely, relevant, and scientifically significant, especially given the urgent need for more sensitive and specific plasma biomarkers in pancreatic cancer.
Despite the clear strengths, the study is limited by a small sample size and lacks independent validation, which restricts the generalizability of the conclusions. However, the technical rigor and innovation make this a valuable contribution to the proteomics and biomarker discovery community.
My specific comments for further improvement:
(1) The pilot study includes only 15 samples (4 controls, 6 primary, 5 metastatic), which severely limits statistical significance. The authors correctly acknowledge this but should include a power analysis or justification for cohort size in the Methods or Discussion.
We thank the reviewer for this valuable suggestion. We have now included an a priori power analysis in the methods on the reliability to detect an effect of at least 2-fold change based on the sample size of the smallest sample group of n=4 (control). Robust statistical analysis (line 171-174) was also performed to ensure detected protein changes within these 15 samples reflect genuine relative differences. This process has been widely applied in high-impact comparative proteomics studies.
The sentence “A priori power analysis indicates that with n=4 per group is sufficient to detect an effect size of >2 fold differences with 80% power, α = 0.05 (CV 50%)” is now included in line 180.
(2) Validation of candidate biomarkers (e.g., ADH1C, LRG1, MDK) via orthogonal methods (e.g., ELISA or targeted MS) is missing. At least a brief outline of planned validation experiments should be added to strengthen translational impact.
Thank you for this constructive and important suggestion. Accordingly, we have been trying to validate the data obtained from proteomic study using Western blot and ELISA. Unfortunately, we have not obtained a clear result from either method due to the difficulty to remove the large amount of background proteins, major serum proteins including albumin, a-antitrypsin, transferin and haptoglobin. We are working on methods to deplete these serum proteins efficiently before we can determine any signaling molecules (proteins) that are less or lowly expressed in the serum. A targeted SRM based assay using heavy labelled peptides is also planned in the future. We have included these planned validation experiments in the discussion (Line 420-424)
(3) The authors propose that extracellular ribosomal proteins could be due to EVs or cancer-related secretion. However, this conclusion remains speculative.
We acknowledge and agree that the presence of ribosomal proteins being EV derived be speculative. We are perplexed on why the supposedly intracellular and abundant ribosomal proteins are higher in individuals with metastatic pancreatic cancer considering lack of any strong and consistent evidence. The first obvious explanation may be due to cell lysis. We then cross referenced them in a plasma contaminant indexes tool, Baize (Gao et al., bioRxiv 2025) to check if they could be derived from platelets, PBMCs, cellular damages or death but couldn’t find any hits. A gene enrichment analysis of all the significantly changed proteins (including the ribosomal proteins) however pointed to enrichment of EVs (Supplementary Figure 6) which have been implicated as carriers of these extracellular ribosomes (Ochkasova et al., Int J Mol Sci 2023).
We have now reworded in the discussion and conclusion that it is a possible enrichment of extracellular vesicles and more work (eg transmission electron microscopy analysis of the nanoparticles) needed to be carried out (Line 459-460). We have also removed the speculative sentence in conclusion on potential enrichment of EV.
(4) Lines 55–57: Add a reference or sentence quantifying the “large dynamic range” of plasma proteins.
We have added the reference by Leigh Anderson and Norman Anderson (MCP, 2002, PMID=12488461)
(5) Table 3: Add Uniprot IDs or accession numbers for completeness.
We thank the reviewer for this observation. The uniport accession numbers are now included in Table 3
(6) Abbreviations: Ensure all are defined upon first use (e.g., DIA, SPD).
We thank the reviewer for this observation. The abbreviations have been expanded on their first use. We have also updated the list of abbreviations used (line 545)
(7) Data availability statement: Confirm that PRIDE accession PDX065103 is active before publication.
Yes, the PRIDE data is currently only viewable by the reviewer using the information below. This will be made public once accepted by the Journal (we need to provide the journal/doi information in PRIDE)
Username: reviewer_pxd065103@ebi.ac.uk
Password: mdxwcw8bk92W
Round 2
Reviewer 2 Report (Previous Reviewer 2)
Comments and Suggestions for Authors
The manuscript has not really been revised, unfortunately.
If it would be about the methodology (Proteonano beads), then the methodology should be explained in much more detail. What is meant by nanoparticles? Nothing else is said but “multi-valent, multi-affinity nanoprobes“. What does it mean? How much them is added to the 80 µl sample? Are the beads coated, how are they coated, how many different kinds of beads (size, coatings, etc.) are being used, and many more details? The new link added to the manuscript in the methods section leads to a webpage in Chinese, even the supposedly English version, and thus does not provide information. Also, the references cited do not provide this data.
In addition, much more about performance had to be revealed. How good is reproducibility, what is the enrichment factor, is it concentration-dependent, does the ratio of particular proteins change upon the enrichment or does it remain identical across the whole spectrum of protein concentration? These and other data would have to be included. Otherwise, there is no scientific content.
If the manuscript would be about biomarker definition, the sample numbers used are totally inadequate. The power calculation only shows that changes could be detected with a reasonable likelihood. It does NOT show that they are relevant. Variation in a single patient or healthy donor could have a strong influence on the result and thus introduce a bias. Therefore, the validity of the potential biomarkers has to be proven in a separate sample set. Otherwise, the biomarker definition is useless. Apart from this, microRNA and protein biomarkers in plasma or serum have been reported that discriminate between healthy and pancreatic cancer with an AUC of 100%. How do the described biomarkers compare to these? The discrimination between “healthy” and “cancer” is not the clinical or biotechnological challenge. The challenges are discrimination between chronic pancreatitis (or other inflammation) and cancer, or between cystic and ductal tumours, or early diagnosis in patients with no apparent risk, or early detection of recurrence after surgical removal.
Unfortunately, the manuscript does not meet either of the above.
This manuscript is a resubmission of an earlier submission. The following is a list of the peer review reports and author responses from that submission.
Round 1
Reviewer 1 Report
Comments and Suggestions for Authors
This study by Ang et al. uses nanoparticle-based plasma enrichment and mass spectrometry to analyze protein changes in pancreatic cancer patients. The authors report deeper protein coverage and identify candidates enriched in metastatic disease. The technical approach is strong, however, the study is limited by a small sample size, lack of validation, and several overinterpreted claims.
Requests to the Authors:
- Validate at least one top marker by another method.
- Remove the claim that the method can help treat cancer at an earlier stage. This is not supported by the data.
- Clarify the clinical stage of the primary tumor group.
- Revise the term “surface marker” in Table 3 to clearly show which proteins are secreted or truly membrane-associated.
- Explain the strong enrichment of ribosomal proteins in more detail.
- Compare the current method to extracellular vesicle proteomics and discuss its advantages or limitations.
- Improve figure labeling, especially in Figure 2C, where text is too small to read.
- Revise the abstract to end with a clearer and more realistic outlook.
- Abstract: remove statements like “potentially” correlate.
- Improve the precision of the language throughout.
- Add a clear statement discussing the limitations of statistical power due to the small sample size, and avoid overinterpretation of group comparisons.
- In the abstract, correct the wording of group descriptions. It should be “patients with primary tumors,” “patients with metastases,” and “healthy controls,” not “tumour” or “tumour that had metastases.”
Recommendation:
Accept with major revisions.
Comments on the Quality of English Language
Grammar is fine, but wording is imprecise.
Reviewer 2 Report
Comments and Suggestions for Authors
In principle, the approach reported in the manuscript is an interesting one. However, the way the analysis was done falls very short of sufficient scientific rigor and provides no innovative information. Technically, nothing new is described; methodologically, a commercial kit was applied, but not checked in terms of what it does to protein abundances or how reproducible it is, for example; clinically, the sample numbers are largely insufficient to make any of the claims made in the paper; biologically, none of the conclusions is any more than speculation since not confirmed otherwise, not even at a basic technical level.
As variations in the process that were used to isolate the proteins could have enormous consequences on the outcome, the reproducibility needs to be documented, in particular if one makes any medically relevant claims. This is missing entirely.
To use merely 4 healthy controls, 6 pancreatic cancer patients with primary tumours and 5 metastatic pancreatic cancer patients is far from sufficient for an appropriate statistical analysis that has relevance to the disease. A substantially larger number of samples needs to be studied!
Also, patients with chronic pancreatitis (or other appropriate controls) have to be included so as to either avoid the very many markers of inflammation or identify and name them as such.
Any identified markers need to be validated in an independent set of appropriate patient samples. Otherwise, they are of no relevance and value!
In conclusion, the paper has several and severe shortcomings that make the reported results very unlikely to be robust and accurate in terms of technical reproducibility, let alone biomedical relevance.